# Atomistic mechanism of phase transformation between topologically close-packed complex intermetallics

Huixin Jin[1,2,3,4,5], Jianxin Zhang [2✉], Pan Li[2,6], Youjian Zhang[2,7], Wenyang Zhang[2], Jingyu Qin [2], Lihua Wang[1], Haibo Long[1], Wei Li[1], Ruiwen Shao[5], En Ma [3✉], Ze Zhang[1,4] & Xiaodong Han [1✉]

Understanding how topologically close-packed phases (TCPs) transform between one another is one of the challenging puzzles in solid-state transformations. Here we use atomic-resolved tools to dissect the transition among TCPs, specifically the μ and P (or σ) phases in nickel-based superalloys. We discover that the P phase originates from intrinsic (110) faulted twin boundaries (FTB), which according to first-principles calculations is of extraordinarily low energy. The FTB sets up a pathway for the diffusional in-flux of the smaller 3d transition metal species, creating a Frank interstitial dislocation loop. The climb of this dislocation, with an unusual Burgers vector that displaces neighboring atoms into the lattice positions of the product phase, accomplishes the structural transformation. Our findings reveal an intrinsic link among these seemingly unrelated TCP configurations, explain the role of internal lattice defects in facilitating the phase transition, and offer useful insight for alloy design that involves different complex phases.

[1] Institute of Microstructure and Property of Advanced Materials, Beijing University of Technology, Beijing 100124, China. [2] School of Materials Science & Engineering, Shandong University, Jinan 250061, China. [3] Center for Alloy Innovation and Design (CAID), State Key Laboratory for Mechanical Behavior of Materials, Xi'an Jiaotong University, Xi'an 710049, China. [4] School of Materials Science & Engineering, Zhejiang University, Hangzhou 310058, China. [5] Beijing Advanced Innovation Center for Intelligent Robots and Systems and Institute of Engineering Medicine, Beijing Institute of Technology, Beijing 100081, China. [6] Institute of Systems Engineering, AMS, PLA, Beijing 100000, China. [7] Shandong Laboratory of Yantai Advanced Materials and Green Manufacturing, Yantai 264006, P. R. China. ✉email: jianxin@sdu.edu.cn; maen@xjtu.edu.cn; xdhan@bjut.edu.cn

Solid-state phase transformations[1–14], accomplished via diffusional or displacive mechanisms, are widely exploited to control the microstructure of metallic alloys. In many cases, the product phase exhibits certain crystallographic relationship with the parent phase[4,5,14,15], rendering the transformation more favorable via specific pathways. A detailed understanding of such structural interrelations and mechanisms has already been reached for many technologically important alloys; well-known examples include the coherent precipitate phases in aluminum alloys[6–11] and martensite formation from austenite in steels[12,13]. However, such a knowledge is sorely missing for the topologically close-packed (TCP) phases[16–19] that frequently appear in advanced alloys such as Ni-based superalloys, the most property-influencing being the rhombohedral μ phase, the orthorhombic P phase, and the tetragonal σ phase. These crystal structures, with unit cell containing up to 56 atoms[16–19], are arguably the most complex end in the spectrum of all TCPs. Yet the intergrowth of P phase or the σ phase in the μ phase is frequently observed, showing sharp and flat interfaces. Our work below aims to explain this poorly understood structural transformation at the atomic scale. We have uncovered an atomic-level mechanism that can effectively establish the new lattice of the product phase from the naturally present planar crystallographic defect in the parent phase, during annealing of the latter at 1100 °C. The presence of planar defects in the parent phases was noticed before[20,21], but was treated as a nuisance obscuring the phase transformation details.

In this work, we use the state-of-the-art high-resolution scanning transmission electron microscopy (HRSTEM) to unveil the role of the defects in the μ phase plays in the intergrowth of the P (or σ) phase. Density functional theory[22–24] (DFT) calculations are used to provide insight into the energetically preferred atomic arrangements constituting the defects. Based on the interrelation between the atomic configurations going from the beginning state to the end state, the transformation between the μ phase and the P (or σ) phase can be construed as a simple dislocation climb mechanism: the diffusional influx of the smaller species inserts 3d transition metal atoms into the excess-space sites at the faulted twin boundary (FTB), resulting in an expanding "extra half plane", accompanied by relaxational rearrangements of the affected neighboring atoms to reconfigure the lattice into that of the new phase. From this vantage point, the transformation mechanism can be construed as one that resembles the climb of a Frank interstitial dislocation loop, but with an unconventional Burgers vector that simultaneously establishes the desired chemical bonds and arrives at the destination configuration of the product phase.

## Results and discussion

**Atomic-resolved structure of planar defects in μ phase.** We start from the μ phase observed in a third-generation superalloy (see supplementary materials for composition details) alloy. Figure 1a shows a high-angle annular dark field (HAADF) image ([1-11] zone axis) and corresponding atomic model of the perfect lattice of the μ phase. The rhombohedral μ phase is found to have lattice parameters of $a = 8.94$ Å, $\alpha = 30.6°$. Out of the large number of μ particles examined, most of them have planar defects inside, as illustrated in detail in Supplementary Fig. 1. The HAADF image of the area containing the defect is shown in Fig. 1b (left panel). From the diffraction spots in Supplementary Fig. 1b, the defect is not a simple (110) twin boundary, nor a simple stacking fault. Instead, it appears to contain both: on the basis of the twinned structure, the region above the twinning plane is displaced by an offset distance (see atomic structural model later). We therefore name this planar defect an FTB.

**Low interfacial energy of (110) defects in μ phase.** We first explain why such FTBs are abundant inside the μ phase. To this end, we carried out first-principles calculations based on DFT. $Co_7W_6$ will be used to simplify Co (Ni, Cr)$_7$W (Mo, Re)$_6$ as the model for the DFT modeling, as our previous work[25] has demonstrated that $Co_7W_6$ can serve as a basis model for the μ phase, a widely reported example of which being Fe(Co)$_7$W(Mo)$_6$[26,27]. To reach a correct model of the defect/interface structure, we first figured out the minimum number ($n$) of atomic layers on either side of the (110) interface required to ensure $Co_7W_6$ bulk properties, by calculating the surface energy of seven different $Co_7W_6$ configurations as a function of $n$, as listed in Supplementary Table 1 (Methods). The $n$ determined as such for each surface configuration is shown in Supplementary Fig. 2, and used to build subsequent defect structures.

Starting from the perfect structure of $Co_7W_6$ in Supplementary Fig. 3a, Supplementary Fig. 3b–i shows the structures with (110) twinning only, marked as T-1~T-8. Several possible types of stacking faults are then added on, to produce the FTB. On the (110) plane, we select two slide directions (shear along [001] for 1/2[001], or along [1-10] for 1/2[1-10]) as representatives. Sliding along [001] for 1/2[001], as shown in the eight interface models in Supplementary Fig. 3b–i, leads to $TS_{[001]}-1 \sim TS_{[001]}-8$ in Supplementary Fig. 4a. Similarly, those formed by sliding along [1-10] for 1/2[1-10] are marked as $TS_{[1-10]}-1 \sim TS_{[1-10]}-8$ in Supplementary Fig. 4a. For comparison purposes, we also calculated the interfacial energy of eight purely twinned structures, and eight single stacking fault structures (Supplementary Figs 3b–i and 4b, c). The interfacial energies of these 32 (110) defects are compared in Supplementary Fig. 5 (Methods). The lowest of all, by far, is the $TS_{[001]}-5$, showing an interfacial energy of only 0.05 J m$^{-2}$. To test the universality of the stability of this defect, the interfacial energies of (110) defects of μ phases with the three constituent components ($Co_7Mo_6$, $Co_7W_3Re_3$, and $Co_7W_3Mo_3$) are also calculated and compared in Supplementary Fig. 6 (Methods). The results indicate that the structures with the minimum interfacial energy in (110) defects of each type of μ phase are all $TS_{[001]}-5$. We therefore conclude that the most stable (110) interfacial termination is of the $TS_{[001]}-5$ type, i.e., (110) twinned parts sliding along [001] for 1/2[001] (Supplementary Fig. 4a). To see if this corresponds to what is observed in experiments, HAADF Z-contrast image simulation is carried out using QSTEM[28], as shown in Fig. 1b (right panel) for $TS_{[001]}-5$; it turns out to be completely consistent (Fig. 1b) with the HAADF image of FTB in experiment. Along [1-10], this composite defect has also been observed, and the HAADF images are also in perfect agreement (Fig. 1c) with the simulated ones. The abundance of planar defects is due to the low stacking fault energy (typically 0.123 J m$^{-2}$) of the TCP[29,30], but more importantly due to extra reduction of energy when the defect is in the form of $TS_{[001]}-5$. This can be explained by noting that pentagonal anti-prism (may be viewed as icosahedra[31] in three-dimensional view) structures (Fig. 1d) are known to stabilize the μ structure[32]. With the configuration of $TS_{[001]}-5$, i.e., the FTB (Fig. 1e), new pentagonal anti-prism structures are created. From this perspective, the FTB reinstitutes the preferred symmetry in the planar defect region, and therefore entails a stability close to that of the perfect μ phase from the structure perspective. This explains why the (110) FTB seen in Fig. 1b is preferred over the twin boundary and so widely formed in the μ phase.

**Intergrowth P phase in μ phase.** Apart from the μ phase, P and σ phase are also common TCP phases in superalloys, presenting obviously different crystal structure[17,18] and the number of atoms in the unit cell. Past experiments have often found σ or P phase

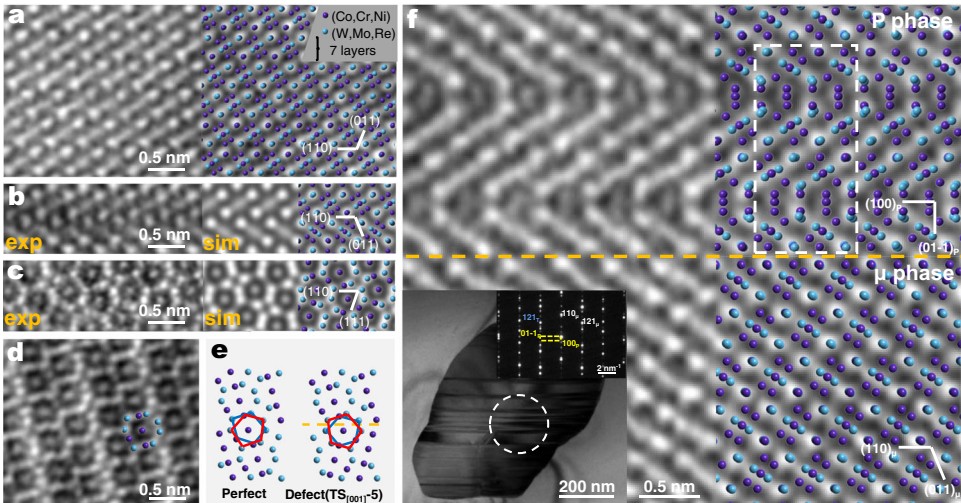

**Fig. 1 The structure and atomic arrangement of the TCP phases and the intrinsic defect, the faulted twin boundary (FTB). a** A HAADF image (left, [1-11] zone axis), in direct comparison with the schematic (right) of μ phase with seven nonequivalent (110) layers. Note that the hexagonal description is also often used for the rhombohedral μ phase. Some typical plane/orientation relationships are: (110) corresponds to (1-12)[H], [1-11] corresponds to [42-1][H], and [1-10] corresponds to [110][H]. Please refer to Supplementary Fig. 1 for details. **b** The left panel displays the experimental HAADF image of the [1-11] projection of (110) FTB; the inset displays the simulated HAADF image together with the schematic of the interface structure $TS_{[001]}$-5. **c** The HAADF image (left) of (110) FTB observed along [1-10]$_μ$ and simulated HAADF images (right) of $TS_{[001]}$-5 along this axis. **d** The HAADF image ([1-10] zone axis) of the μ phase, showing pentagonal configurations. **e** The projected atomic arrangement of μ phase and the $TS_{[001]}$-5 along [1-10]$_μ$ in which the pentagonal anti-prism structures are highlighted. **f** The HAADF image and the schematic of the P phase and μ phase, projected along [011]$_P$ and [1-11]$_μ$, respectively; the boundary plane marked by orange dotted line between the two phases is (110)$_μ$, parallel to (100)$_P$. The white framed box marks the unit cell of the P phase. The lower left corner inset displays the bright-field TEM micrograph and the corresponding SAED patterns of the two phases.

growing inside the μ phase[20,21]. We also frequently observe the intergrowth P phase in μ, as indicated by the diffraction pattern given in the inset (lower left corner) in Fig. 1f, which displays the HAADF image and the atomic schematic showing the P phase and μ phase, projected along [011]$_P$ and [1-11]$_μ$, respectively; The crystallographic information[17] of each phase and their orientational relationships[20,21] are listed in Supplementary Table 2. We observe that the interface between the two phases is flat, with very regular atom arrangement and a high degree of coherency, implying a possible configurational connection between these two seemingly unrelated structures. We note that FTBs can be found in the vicinity of all intergrowth P phases in our samples. This is an indication that the phase transformation between μ and P may be related to the (110) FTB. We will see in the following how the FTB bridges the two structures, upon the insertion/in-flux of the extra 3d transition metal atoms.

**Atomic-resolved structural transformation between μ and P (or σ) phases.** With the above atomic-level view of the parent structure and the product structure at the FTB, we now proceed to picture how the conversion from the parent μ lattice into the product P (or σ) lattice can be accomplished. Figure 2a–d schematically shows a process that can convert the μ structure, facilitated by the (110) FTB, into a 15-layer slab that has the same atomic packing inside as in the structure of the P (or σ) phase. As will be explained later, to form the P (or σ) structure with complete periodicity, two such slabs need to be formed at the same time from two FTBs that are close by; which phase actually forms, either P or σ, depends on the spacing between the two FTBs. Figure 2e–g depicts more details involved via this structural transition route. Specifically, Fig. 2e displays the structure at the original FTB, in the [1-11] projection. As there is extra space (volume per atom being larger than elsewhere) between the seven-layer slabs (see, e.g., the visible "gap" in Fig. 1b), a diffusion pathway is available for the smaller 3d metal atoms to diffuse in,

as schematically shown in Fig. 2f, taking the excess space at the FTB of the parent μ lattice. This in-flux creates an extra plane of atoms, which, much like a Frank interstitial dislocation loop, climbs to the right as more 3d metal atoms arrive. This would incur considerable lattice expansion/strain on the neighboring atoms as well as a need to adjust chemical bonds; the synergistic relaxational displacement is schematically shown in Fig. 2c. Different from the conventional Frank dislocation loop, here the climbing dislocation involves a Burgers vector that has both a horizontal component and a vertical component. As depicted in Fig. 2f, when this gradual climb process is over for the entire orange dotted box, the atoms inside the box would have all been relocated, each having experienced a translational displacement with Burgers vector $\mathbf{b}_{\parallel} = 1/2[1\text{-}10]$ (the double-ended arrow marks its [1-11]$_μ$ projection) and a vertical displacement with $\mathbf{b}_{\perp}$ (its magnitude is 1.4 Å), resulting in the scenario in Fig. 2g after simple relaxation. Note here that this mechanistic picture is borne out of experimental evidence: the HAADF images in Fig. 2h–j, recorded in our experimental observations, directly correspond to the respective atomic model projections in Fig. 2e–g. Note that the atomic configuration inside the gray dotted box seen in Fig. 2g turns out to be a typical portion of the P (or σ) phase structure (compare with Fig. 1f). In other words, diffusional climb having an unconventional Burgers vector produces the configurational relationship depicted in the scenarios in Fig. 2e–g, and the end result is that the parent μ lattice with FTB would transform through this mechanism into the partial structure of P (or σ) phase. The overall process could be simplified using the cartoons in Fig. 2a–d.

To nail down the above mechanism in action, we examine the TEM snapshots of the intergrowth front, which do support the picture in Fig. 2. Figure 3 is the HAADF image capturing a particular moment, encompassing three zones (Boxes A, B and C) representing different stages in the phase transformation (propagating to the right, with the yellow 3d atoms in the interstices at the original FTB of the μ phase). Box A shows a

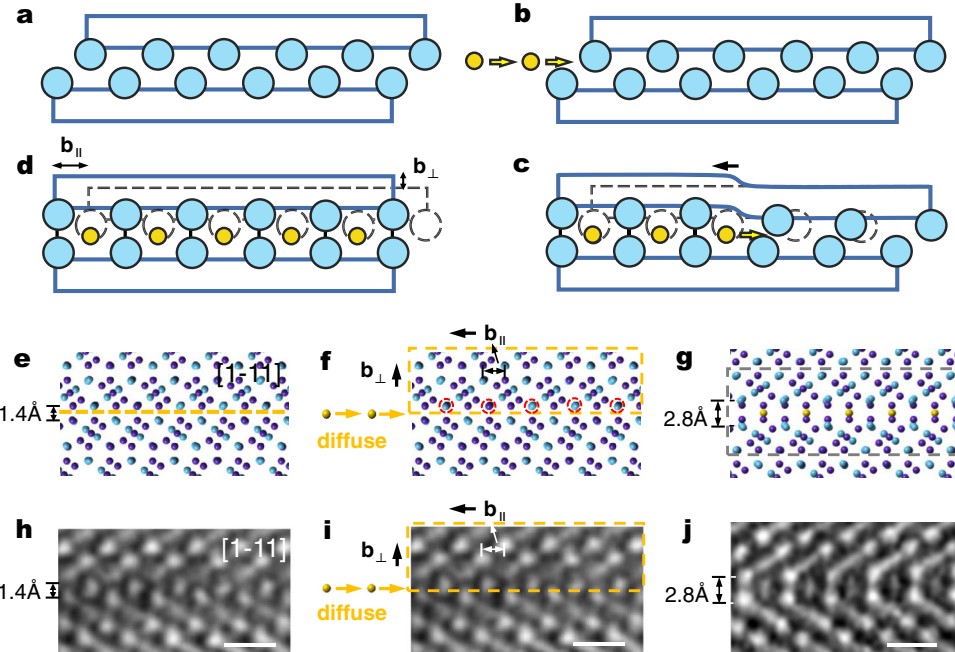

**Fig. 2 The transformation path from the (110) FTB of μ phase to the partial structure of P (or σ) phase.** Note that the hexagonal description is also often used for the rhombohedral μ phase. Some typical plane/orientation relationships are: (110) corresponds to (1-12)H, [1-11] corresponds to [42-1]H, and [1-10] corresponds to [110]H. Please refer to Supplementary Fig. 1 for details. **a–d** Schematic illustration of the transformation process via dislocation climb. **a** The original FTB of the μ phase. **b** Smaller metal (yellow) atoms diffuse into the FTB. **c** The resultant dislocation, resembling a Frank interstitial loop, climbs to the right, converting the local structure into that of the P phase, which is shown in (**d**). Note that the Burgers vector of the climbing dislocation has both a parallel and a vertical component. **e–j** illustrate the details using an atomic model. The purple and blue atoms represent (Co, Cr, Ni) and (W, Mo, Re), respectively. **e** Atomic model of the FTB, projected along [1-11]$_\mu$. The orange dotted line marks the interface position of the FTB, into which the small atoms (represented by yellow balls) in (**f**) diffuse/insert. **f** An extra row of atoms gradually diffuse into the FTB; see Supplementary Fig. 8 for additional information and red dashed circles. All the atoms inside the orange dotted box experience translational displacement with Burgers vector $\mathbf{b}_\parallel = 1/2[1\text{-}10]$ and vertical displacement with $\mathbf{b}_\perp$ (its magnitude is 1.4 Å), resulting in (**g**), a partial structure of P phase. **h–j** The HAADF images corresponding to the atomic model projections in (**e–g**). Scale bars of (**h–j**) are 0.5 nm.

region where the atomic arrangement in this location is already that of P, i.e., the conversion is over and all distortions have relaxed. The tip of the transformation zone is now reaching Box B, in which the lattice distortion strain is still discernible. To its right, Box C (meant to be a representative untransformed region) still remains FTB with little distortion, at a location where the transition to P has not yet started. We can deduce from this sequence of events that the existing (110) FTB of the μ phase sets the stage for the transformation to the P phase. The structural change is triggered by an extra row of 3d metal atoms (mainly Cr, and the HAADF mapping indicates that Cr is indeed enriched here after the transformation, as seen in Supplementary Fig. 7) diffusing into the interstices at the FTB, necessitating atomic rearrangements. As seen in Fig. 3, the structural transformation front is currently at the boundary between Box B and Box C. One can envision that the climbing dislocation (extra half plane composed of 3d atoms) has reached this location (compare with Fig. 2c). When fed by the in-flux of more and more smaller 3d transition metal atoms, further climb would expand Box C, extending the front of the intergrowth P slab. Note again that the atoms inside Box C would undergo not only a vertical displacement, but also a horizontal displacement, to reenact the scenario in Box B. These displacements have been depicted using the vertical and horizontal components of a Burgers vector in Fig. 2. This Burgers vector (red) can be readily identified in Fig. 3b, where a Burgers circuit is drawn to enclose the climbing dislocation.

**(110) FTBs of μ phase determine the phase transformation.** The transformation process above can result in either the P or the σ phase—it is the spatial location/distribution of the FTBs that determines the type and size of the new phase formed from the parent μ structure. This is illustrated in Fig. 4. Figure 4a is the projected atomic arrangement of μ phase with one FTB in which the FTB is marked with orange dotted line, both sides of the FTB are the perfect structure of μ phase with seven nonequivalent (110) layers. The thickness of the seven layers is defined as block D. In Fig. 4d, two FTBs are present, apart by spacing D. In this scenario, the transformation (as discussed in Figs. 2 and 3 and Supplementary Fig. 8) proceeds along both FTBs, as shown in Fig. 4e, where the region inside the orange dotted box has formed the complete structure of the σ phase (compare the simulated and the observed images for the σ phase, Fig. 4f). When the spacing between the two FTBs is 2D, as depicted in Fig. 4g, the structural transformation along both FTBs would complete the periodicity required of the P phase (compare the simulated and the observed images for the P phase, Fig. 4i), as shown in Fig. 4h.

It can be seen from the above that the formation of regular new P (or σ) phase requires a specific spacing between FTBs in μ phase, while the initial distribution and spacing of FTB in μ phase are random to some extent (Fig. 1f and Supplementary Fig. 9b, c). We will discuss below how the new phase thickens through the short-range diffusion of local atoms at the FTBs. Figure 5a is the HAADF image captured at a specific moment in the thickening process of intergrowth P phase, which includes three zones (A, B, and C),

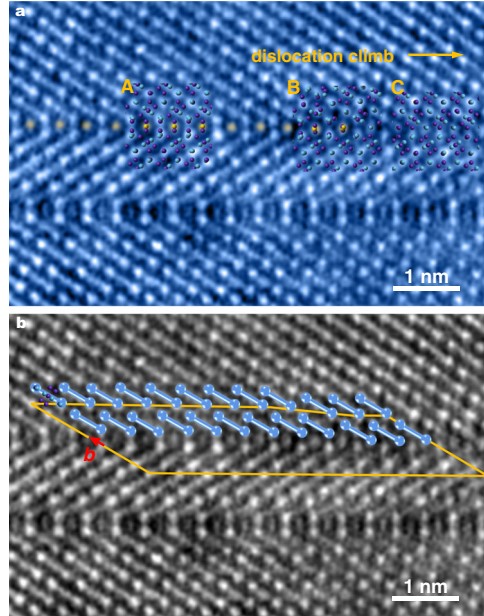

**Fig. 3 The HAADF image projection along [1-11]$_\mu$ showing the transformation from (110) FTB of μ phase to P (or σ) phase in progress. a** is lightly colored to enhance brightness and contrast. The positions of the diffusing-in atoms during phase transformation are colored orange. Boxes A–C in (**a**) correspond to the three sequential stages that catch the phase transformation in progress; At location A, the transformation to the partial structure of P (or σ) phase is completed. Location B captures a snapshot of the core of the climbing dislocation, i.e., the boundary zone between A and C (the original (110) FTB of the μ phase). The orange arrow marks the direction of both the dislocation climb and the progressive phase transformation. The purple and blue atoms represent (Co, Cr, Ni) and (W, Mo, Re), respectively. **b** illustrates the Burgers vector circuit of the climbing dislocation. The dumb-bell symbol represents a structure unit containing the atoms in seven nonequivalent (110) layers of the μ phase. The orange Burgers circuit fails to close (compare with the closed Burgers circuit of FTB in Supplementary Fig. 9a), and the red arrow shows the Burgers vector that can be decomposed into a horizontal and vertical component, and they are consistent with $\mathbf{b}_\parallel$ and $\mathbf{b}_\perp$ in Fig. 2.

representing different stages of transformation (propagating to the upper right along the orange arrow). The arrangement of atoms in region A (containing thicker and more P phases) is shown in Fig. 5b, where the conversion is over and all distortions have relaxed. The tip of the transformation zone is now reaching region B (transition zone), in which the lattice distortion strain is still discernible. At the upper right, the atoms in region C (containing thinner and fewer P Phases) are arranged as shown in Fig. 5c, and its transformation to region A, has not yet begun. Comparing the atomic arrangement of region A and region C, it can be found that, except for some atoms in the blue block layer (thickness of 4D) of region A, the arrangements of the atoms on both sides above and below are exactly the same as region C. Also, in terms of the atom numbers and the relative positions between atoms, the blue block layer is consistent with the same block layer of region C. We can deduce from this sequence of events that the transformation from region C to region A, i.e., the thickening process of intergrowth P phase can be accomplished by the short-range diffusion (shuffle) of local atoms, and it is similar to a horizontal flip from the planar view. This transformation changed the distribution and spacing among FTBs (shown by black dotted line in Fig. 5b, c), and the number of FTBs with a spacing of 2D increased significantly after the transformation. Figure 4 shows that the regular P phase requires a spacing of 2D between FTBs, so the transformation from

region C to region A thickens the P phase. The transformation process can be further simplified to Fig. 5d, and the atomic structures of region C and A are simplified as line 1 and line 2, respectively. Endpoints and turning points of line 1 and line 2 indicate the presence of the FTBs here. The local atomic shuffle performs a transformation from the middle red line to the orange line, and the transformation can change the location of FTBs (The FTB at number 5 transforms to perfect μ and an identical FTB appears at number 5'). When the same transformation happens to line 3 (Fig. 5e), which has similar structure to line 1, it can change the number of FTBs (from four to six). To illustrate, Fig. 5f is used to describe the growth process of the intergrowth P (or σ) phase. During the formation of the parent μ phase, there will be a lot of initial randomly distributed (110) FTBs with low stacking fault energy, which is represented by the solid black line in schematic (i), with a total number of m. When a new phase is more stable at a certain heat treatment temperature, via a similar local atomic short-range diffusion at FTBs like Fig. 5a, the number of FTBs n can proliferate to 3n, as shown in schematic (ii) of Fig. 5f, and their distribution and spacing can also be regulated as schematic (iii) in Fig. 5f. The FTBs with certain spacing required for the growth of new phases can be thus obtained by this short-range diffusion processes, as shown in Figs. 4 and 5. The driving forces of this transformation are the difference in Gibbs free energy of these phases at the heat treatment temperature.

In summary, we have shown that even though the TCP structures appear to be complex and vastly different[17,18], there is a fairly straight-forward pathway for the conversions between the μ and P (or between the μ and σ) structures. The transformation is mediated by the long-range diffusion of the smaller 3d transition metal atoms into the relatively more spacious (open-structured) FTB, synergistically accompanied by local atomic rearrangements that adjust the interatomic/interplanar spacing at the intergrowth tip. One can view the mechanism from a "dislocation climb" perspective: The extra half plane composed of 3d atoms at the FTB forms a dislocation that resembles a Frank interstitial loop. This dislocation climbs, with a Burgers vector having both a horizontal component and a vertical component; together they nudge the neighboring larger atoms into new lattice positions—the desired atomic locations dictated by the chemistry and symmetry of the product phase. Our findings link together the seemingly unrelated TCPs: they are bridged by the FTB and fast-diffusing smaller atoms. The insight also sheds light on the planar morphology of the intergrowth phases, as well as the diffusion-limited TCP transformation kinetics in general. Furthermore, the Frank partial dislocations similar to that in our work have recently been reported[33] in MAX phase materials, which may play an important role in the field of self-lubrication, oxidation-resistant coatings, self-healing and energy materials. These defects were found to have close relations with the phase transformation between the materials. Our work is expected to promote the understanding of structural transformations within these and other similarly complex materials. Additional abstract and/or mathematical treatment, by taking for example the standpoint of the evolution of disconnections at the interphase boundaries[34,35], may formulate the phase transformation in a more analytical manner, but is beyond the scope of this work.

## Methods

**Preparation of samples**. The experimental alloy was solidified directionally in order to obtain single crystal bars with dimensions of 14 mm × 150 mm along the [001] growth orientation. Its nominal composition is Ni-6Co-5Cr-3Mo-6W-5Re-6Al-5Ta (wt%). The bar went through a series of heat treatment, including heating to 1300 °C for 1 h and 1340 °C for 5 h, followed by air cooling, and then at 1100 °C for 4 h and 870 °C for 20 h, followed by air cooling. The final ageing was done at 1100 °C for 1000 h to promote the formation of TCP phases.

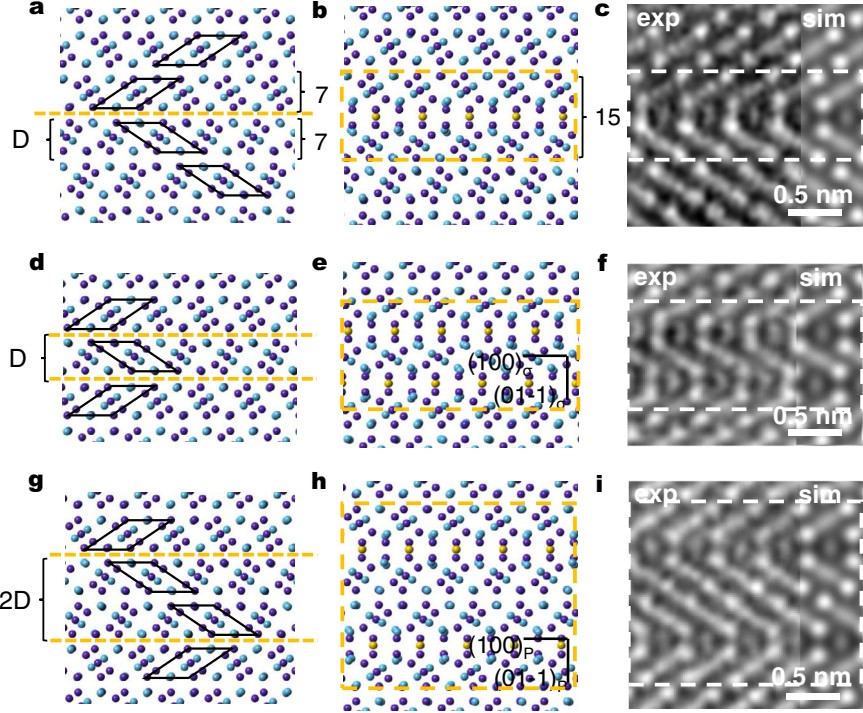

**Fig. 4 The transformation from μ can lead to either the P or the σ phase. a** The projected atomic arrangement of μ phase with one FTB; the FTB is marked with orange dotted line, both sides of the FTB are the perfect structure of μ phase with seven nonequivalent (110) layers. The thickness of the seven layers is defined as block D. The purple and blue atoms represent (Co, Cr, Ni) and (W, Mo, Re), respectively. After the transformation displayed in Figs. 2 and 3, **a** changes to the partial structure of P (or σ) phase (marked with framed boxes) in (**b, c**). The yellow atoms are the extra row of atoms that have diffused in. **d** The atomic arrangement showing two FTBs of μ phase with a spacing of D (between the two orange dashed lines). **e, f** The model and corresponding experimental HAADF image of defects in (**d**) after transformation, and the part in the dashed boxes are the complete intergrowth σ phase projected along [011]σ. **g** The atomic arrangement of two FTBs of μ phase with a spacing of 2D. **h, i** The model and corresponding HAADF image of defects in (**g**) after transformation, where the region inside the dashed boxes are the complete intergrowth P phase projected along [011]P. The HAADF image simulation of each structures are included as overlaid insets in (**c, f, i**), respectively.

**TEM and HRSTEM characterizations.** Phase identification and morphology observation of TCP phases were carried out using TEM. HRSTEM was used to observe the structures of different TCP phases and their internal composite defects at atomic level and capture a snapshot of the core of the climbing dislocation during the phase transformation process.

Tecnai T20 TEM, which has a double tilt sample holder for structural analysis, was set at 200 kV. Atomic resolution, Z-contrast (caused by the differences in atomic numbers), imaging [through HAADF-HRSTEM] was performed in a FEI Titan Themis spherical aberration-corrected microscope operating at 300 kV.

The samples were cut to 300-μm-thick slices via electrical discharge machining, and polished to 80 μm with silicon carbide paper. The specimens with 3 mm diameter were punched out mechanically and grounded to around 25 μm in thickness. Using an electrolyte solution of 10% perchloric acid and 90% ethyl alcohol, the foils' thicknesses were further reduced for TEM observations through twin-jet electrochemical polishing at −20 °C. The HAADF-STEM image simulations were carried out via QSTEM software[28].

**Parameter setting of DFT calculations.** All structural relaxation and energetic calculations were done via the DFT code - VASP (Vienna Ab Initio Simulation Package)[22] with periodic conditions and the plane-wave basis sets[23]. The interactions between electron ions were handled by the method of projector-augmented wave theory[22–24]. Generalized gradient approximation method was utilized to describe exchange-correlation energy[36,37]. The cut-off energy was set to 400 eV in all calculations. K-point sampling grids obtained by Monkhorst–Pack method[38] were set to $10 \times 10 \times 10$ for bulk $Co_7W_6$, and $10 \times 6 \times 2$ for all the slabs and interfaces. The vacuum layer was 10 Å in thickness in the surface and interface model. Broyden Fletcher Goldfarb Shannon algorithm was utilized to relax the models and optimize the structures. The energy changes in the structural optimization process and the maximum stress converge to $1.0 \times 10^{-5}$ eV/atom and 0.01 eV/A, respectively. To make it easier for readers to track the research results, the Wyckoff positions of rhombohedral μ, hexagonal μ, σ and P phases are provided in Extended Data Table 2. Also, the cif files of these phases, FTBs with different spacings, and the structures of these FTBs after the transformation are put in Supplementary Data 1–9.

**Calculation of the (110) surface energy in μ phase.** The (110) surface of $Co_7W_6$ is a polar one, and there should be seven different surface termination types depending on the packing of the surface atoms (Supplementary Fig. 2), marked as Type 1–Type 7. The surface energy σ of $Co_7W_6(110)$ can be calculated as[39]:

$$\sigma_{Co_7W_6(110)} = \frac{1}{2A}(E_{slab} - N_{Co}\mu_{Co} - N_W\mu_W) \qquad (1)$$

Here, $E_{slab}$ is the total energy of the surface structure; $N_{Co}$, $N_W$, $\mu_{Co}$, $\mu_W$ are the number of atoms and corresponding chemical potential of Co and W, respectively; $A$ is the surface area.

The chemical potential of the inner atoms of the interface models under equilibrium should satisfy the following relationship:

$$\mu_{Co_7W_6}^{pure} = 7\mu_{Co} + 6\mu_W \qquad (2)$$

Herein, $\mu_{Co_7W_6}^{pure}$ is the energy of pure $Co_7W_6$.

Combining the formula above,

$$\sigma_{Co_7W_6(110)} = \frac{1}{2A}(E_{slab} - N_{Co}\mu_{Co} - N_W\frac{\mu_{Co_7W_6}^{pure} - 7\mu_{Co}}{6}) \qquad (3)$$

Our past work[25] have revealed that the difference between $\mu_{Co}$ and the chemical potential of bulk Co is too small to notice. That is to say, making $\mu_{Co} = \mu_{Co}^{pure}$ will make no difference in the following calculation results. The calculated $\mu_{Co_7W_6}^{pure}$ and $\mu_{Co}^{pure}$ are −128.21 and −7.11 eV, respectively. From formula (3) the surface energies of the seven different $Co_7W_6$ (110) are listed in Supplementary Table 1. As can be seen, for each case, with increasing number of atomic layers the surface energy gradually approaches convergence, such that each surface model is able to reflect bulk characteristics.

**Calculation of interfacial energy of defects.** The work of separation $W_{sep}$ and interfacial energy $\gamma_i$ are closely related to interface structures and characteristics, which can be used to estimate the bonding strength and the stability of the interface defects. $W_{sep}$ is defined as the necessary reversible work to separate an interface

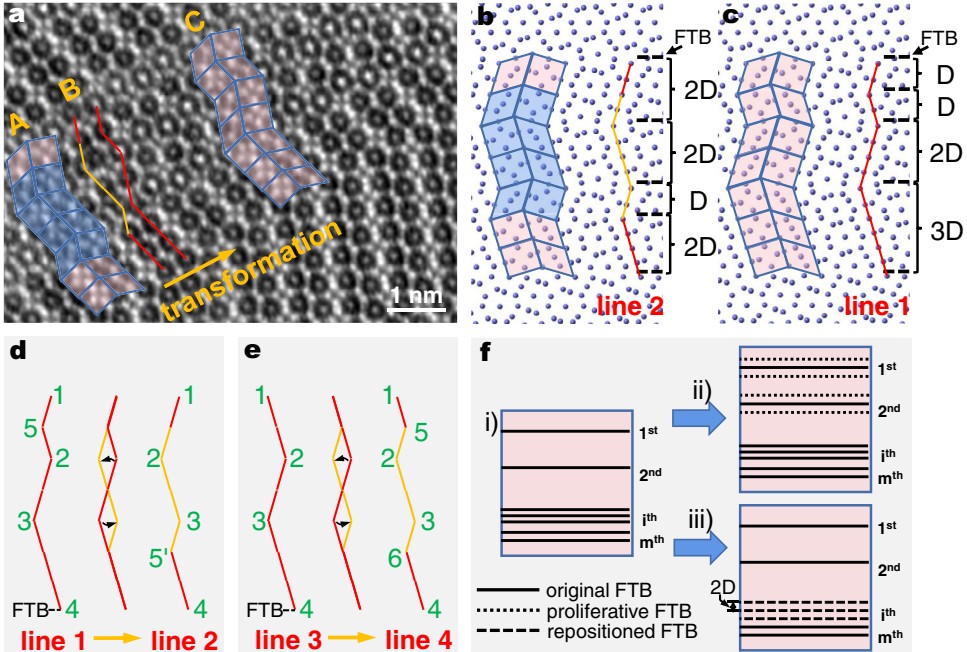

**Fig. 5 The HAADF image and schematic illustration of the thickening process of intergrowth P phase. a** The HAADF image projection along $[1\text{-}10]_\mu$ showing the thickening process of intergrowth P phase. The orange arrow in (**a**) marks the direction of the progressive transformation. Regions A–C corresponds to the three sequential stages of transformation. **b, c** Atomic models corresponding to the regions A and C in (**a**), respectively. The differences between the atomic arrangement in region A and that in region C are indicated by the blue block layer (thickness of 4D) in (**b**). The black dashed lines mark the interface position of the FTBs in (**b, c**) and the distances between two adjacent FTBs are added. The atomic structures of region C and A can also be simplified as line 1 and line 2, respectively. Their endpoints and turning points indicate the presence of the FTBs here, numbered by the green text in (**d**). **d** Schematic illustration of the location of FTBs changed by the transformation in (**a**). The FTB at number 5 transforms to perfect μ and an identical FTB appears at number 5′. **e** The schematic example of the number of FTBs changed by the same transformation. The structure of line 3 is similar to that of line 1, only different at the top. **f** The brief schematic of the growth process of intergrowth P (or σ) phase in μ phase. The schematic (i) in (**f**) shows that there are many randomly distributed original (110) FTBs in the parent μ phase, represented by the solid black lines, with a total number of m. Via the local atomic short-range diffusion at FTBs, n FTBs can proliferate to be 3n as illustrated by the first and second FTB shown in (ii), and their distribution and spacing can also be regulated as illustrated by the ith and nearby FTBs shown in (iii). The new phase in schematic (iii) is an example of the P phase (FTBs with a spacing of 2D). If the new phase is σ phase, the spacing between FTBs is D.

into two free surfaces, as it can be calculated via the following formula[40,41]:

$$W_{sep} = \frac{1}{A}(E_P + E_T - E_{P/T}) \qquad (4)$$

$E_{P/T}$ is the total energy of the interface structure; $A$ is the interface area; $E_P$ is the laminate energy of $Co_7W_6$ (110) with perfect structure; correspondingly, $E_T$ is the laminate energy of $Co_7W_6$ (110) with twinning structure.

The $W_{sep}$ values of 32 possible interface structures of the (110) defects have been calculated and listed in Supplementary Table 3. It can be demonstrated that the work of separation of $TS_{[001]}$-5 is the largest, so it can be reasonably speculated that the interface atomic bonding strength of this model is the strongest. Based on this, we have calculated $\gamma_i$ of the 32 interfacial terminations. $\gamma_i$ is the important parameter to evaluate interface stability, because the smaller its value, the more stable the interface structure. $\gamma_i$ can be calculated as[40]:

$$\gamma_i = \sigma_P + \sigma_T - W_{sep} \qquad (5)$$

where $\sigma_P$ and $\sigma_T$ are the surface energy of the laminates with perfect and twinning structure on both sides of the (110) defect, respectively. See $\gamma_i$ for each termination of the (110) defects, as displayed in Supplementary Fig. 5.

Moreover, $\gamma_i$ of (110) defects of μ phases with three constituent components ($Co_7Mo_6$, $Co_7W_3Re_3$, and $Co_7W_3Mo_3$) have been calculated using the above methods. $\gamma_i$ of 32 (110) defects of each type of μ phase are shown in Supplementary Fig. 6.

## Data availability

All data generated in this study are provided in the Supplementary Information.

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

## Acknowledgements

NSFC programs (91860202, 52071003, 11604006, 51771102, 51971118, 52171001). The authors at BJUT wish to acknowledge the financial supports of Basic Science Center Program for Multiphase Evolution in Hyper-gravity of the National Natural Science Foundation of China (51988101), Beijing Municipal Education Commission Project (PXM2020_014204_000021 and PXM2019_014204_500032), Beijing Outstanding Young Scientists Projects (BJJWZYJH01201910005018), Beijing Natural Science Foundation (Z180014), and the "111"project (DB18015). E.M. acknowledges XJTU for supporting his work at CAID. The authors acknowledge Analysis and Testing Center in Beijing Institute of Technology.

## Author contributions

J.X.Z. and X.D.H. conceived and supervised the project. H.X.J. made the samples, conducted the TEM experiments and first-principles calculations with the help of P.L., Y.J.Z., W.Y.Z., J.Y.Q., L.H.W., H.B. L., W.L. and R.W.S. under the supervision of J.X.Z., Z.Z. and X.D.H. H.X.J. prepared the figures under the supervision of X.D.H. E.M., H.X.J. and X.D.H. led the interpretation of the results. All authors contributed to the data analysis and discussions.

## Competing interests

The authors declare no competing interests.
