## [Peer Review File · Nature Communications]

Title: Atomistic Mechanism of Phase Transformation between Topologically Close-packed Complex IntermetallicsREVIEWER COMMENTS

Reviewer #1 (Remarks to the Author):

The work is interesting, the results sound, possibly important, references, language and most figures appear fine, but it is a hard read also for experts as it deals with complex geometry.

The main problem is the presentation. I am not unfamiliar with complex structures, but had to use much time to try to visualise the results. I feel this should not be necessary. Even structures with more atoms are possible to present in simpler ways. The question is of cause, how.

My opinion is that coordinates of the structures (μ , σ and P) should be given, if possible, such as in the extended data part, to allow readers (with suitable software) to follow the findings. The use of the rhombohedral space group instead of the hexagonal (for the μ -phase) is correct, but the crystallographic directions are much harder to visualise as rhombohedral, (although the authors do show both cells in the extended parts). I personally would prefer to have coordinates in both space groups be given.

There are few errors in the text, but one that is repeated several times, unless there is a new method I don't know of, called HADDF, I suggest it be HAADF.

On line 45, the authors refer to similarities with the coherent precipitates in Al alloys, which is fine. However, I am not sure if the references serve the intended purpose. I would be more impressed if they had detected the structural relations that exist between the μ -phase and the precipitate phases in the Al 7xxx system. Yes, there are such relations.

The statement in line 50 that these phases are more complex than Laves phases, I cannot fully agree on. Laves phases, although the structure of phases like $Zn_{2}Mg$ are relatively simple, they do display complex stacking faults (as their precursors in the 7xxx system alloys clearly show), and they are structurally related through (disordered) FK phases to quasicrystals, which are what you would call 'more complex' than these phases.

Line 52 claims 'unusually flat and sharp interfaces'. What is 'unusual'? Are the stacking faults flatter and sharper than in diamond silicon? I think not. Neither Si nor stacking faults in Si are unusual. The claim is wrong.

The figures and annotation are ok, except they could be simplified.

In short: This work has sound and interesting findings, well documented. The findings relate to how TCP phases in super-alloys connect structurally. The findings may be important, as these alloys are technologically materials of increasing importance. However, the presentation is very technical. Unless I

misunderstand the directions of Nature Communications, my opinion is that this work fits better in specialist journals like Acta Materialia.

A final comment:

In Fig. 1e, the authors describe a motif as 'pentagonal' configurations. These are the same that exist with precipitates in the 7xxx system, and may be viewed as centred deformed icosahedra.

Reviewer #2 (Remarks to the Author):

In this manuscript, the authors studied the atomic arrangement of the faulted twin boundary (FTB) in the μ phase and the intergrowth of P (or σ) phase with μ phase at the FTB by HRSTEM. It is proposed that the FTB in the μ phase sets up an easy pathway for the long-range diffusional transport of the smaller 3d transition metal species and the affected neighboring atoms rearrange to reconfigure the lattice into that of the P (or σ) phase. This study provides useful information about the planar defects and the mechanism of the intergrowth of P (or σ) phase with μ phase. However, since the stacking fault energy can be significantly influenced by the element and composition, the energies of the planar defects calculated by using a simplified Co7W6 model are not convincing. The reason why the spatial location/distribution of the FTBs instead of thermodynamics determines the type and size of the new phase formed from the parent μ structure has not been properly discussed. The detailed comments and questions from the reviewer are as follows.

1. The authors reported that FTBs are abundant in the μ phase. According to literature [1] different types of planar defects on the $\{1 -1 0 2\}$ plane were observed in μ phase. Do other types of non-basal planar defects exist in the μ phase in the present work? Are the FTBs the dominant planar defects in the sample? How are the FTBs distributed? Does the intergrowth of P (or σ) phase with μ phase only occur at the vicinity of the FTBs? The information is necessary for readers to understand why the authors correlated the intergrowth of P (or σ) phase with the FTBs rather than the other planar defects.

[1] S. Gao, Z.-Q. Liu, C.-F. Li, Y. Zhou, T. Jin, In situ TEM investigation on the precipitation behavior of μ phase in Ni-base single crystal superalloys, Acta Materialia 110 (2016) 268-275.

2. The authors used Co7W6 as a simplified model to calculate the interfacial energies of several non-basal defects and concluded that the FTBs in the form of TS[001]-5 are abundant because of the lowest energy. However, in literature [2] experimental results show that there are various types of non-basal planar defects in Co7W6 but the FTB in the form of TS[001]-5 was not reported. It indicates that at least the FTB in the form of TS[001]-5 is not dominant in Co7W6. Therefore, it is not convincing to explain why such FTBs are abundant in the μ phase using the model of Co7W6. Besides, such FTBs were not frequently observed in other superalloys having different compositions. Since the element and chemical composition can influence the stacking fault energy, the occurrence of this type of defect might be related to the composition of the sample studied in the present work. Have the authors considered the influence of the element and chemical composition on the defects?

[2] P.A. Carvalho, J.T.M. De Hosson, Stacking faults in the Co7W6 isomorph of the μ phase, Scripta

Materialia 45(3) (2001) 333-340.

3. The authors concluded that “it is the spatial location/distribution of the FTBs that determines the type and size of the new phase formed from the parent μ structure”. The P and σ phases are expected to have different thermodynamic stabilities at the heat treatment temperature, and thus the driving forces for the phase transformations from the parent μ to P and from μ to σ are expected to be different. How can the spatial location and distribution of the FTBs overwhelm the thermodynamics and control the phase transformations? Moreover, if the P and σ phases are thermodynamically more stable than the μ phase at the heat treatment temperature, they are expected to grow at the expense of the μ phase [3]. If the size of the new P and σ phases are also determined by the spatial location/distribution of the FTBs, how do the P and σ phases grow?

[3] X.P. Tan, J.L. Liu, T. Jin, Z.Q. Hu, H.U. Hong, B.G. Choi, I.S. Kim, C.Y. Jo, Intergrowth of P phase with μ phase in a Ru-containing single-crystal Ni-based superalloy, *Philosophical Magazine Letters* 92(10) (2012) 556-562.

Reviewer #3 (Remarks to the Author):

In their manuscript “Atomistic Mechanism of Phase Transformation between Topologically Close-packed Complex Intermetallics”, Jin et al. present an in-depth experimental and numerical study of the formation and transformation of topologically close-packed (TCP) phase. By focusing on Ni-base superalloy materials, the authors demonstrate the transformation of a μ phase into other TCP phases (P or σ) by means of the climb of a Frank partial dislocation.

The manuscript presents very precise atomic-scale experimental evidences of the formation and propagation of the Frank dislocation. I particularly appreciate the demonstration of the unconventional Burgers vector. The first principles calculations supporting the experiments are judiciously performed on a surrogate material as the complete superalloy remains obviously out of range of the capabilities of the DFT method. These calculations provide a strong support in building the defect structures presented in Fig. 2 and Fig. 4.

The mechanism evidenced by the authors is highly interesting for the community and is probably transferable to other class of material. As an example, it has been recently observed Frank partial dislocations in Ti-base MAX phases [Yu et al, *Scripta Materialia* 191 (2021) 34] and this defect could be associated with phase transformation by following a mechanism similar as the one proposed by this work. A short comment on the transferability into other TCP / complex phases of the mechanism evidenced by the authors could strengthen the discussion of the manuscript.

Reviewer #1 (Remarks to the Author):

The work is interesting, the results sound, possibly important, references, language and most figures appear fine, but it is a hard read also for experts as it deals with complex geometry.

The main problem is the presentation. I am not unfamiliar with complex structures, but had to use much time to try to visualise the results. I feel this should not be necessary. Even structures with more atoms are possible to present in simpler ways. The question is of cause, how.

My opinion is that coordinates of the structures (μ , σ and P) should be given, if possible, such as in the extended data part, to allow readers (with suitable software) to follow the findings. The use of the rhombohedral space group instead of the hexagonal (for the μ -phase) is correct, but the crystallographic directions are much harder to visualise as rhombohedral, (although the authors do show both cells in the extended parts). I personally would prefer to have coordinates in both space groups be given.

Response: We thank the reviewer for these suggestions. We do agree that providing coordinates of the structures make it easier for readers to track and visualize the results. According to these advices, we've provided the Wyckoff positions of rhombohedral μ , hexagonal μ , σ and P phases in Extended Data Table 2. Also, the cif files of these phases, FTBs with different spacings, and the structures of these FTBs after the transformation are put in Supplementary Data 1-9. These can help readers track the research results with suitable software.

There are few errors in the text, but one that is repeated several times, unless there is a new method I don't know of, called HADDF, I suggest it be HAADF.

Response: We are very sorry for the typo, and they have been corrected in the revised manuscript.

On line 45, the authors refer to similarities with the coherent precipitates in Al alloys, which is fine. However, I am not sure if the references serve the intended purpose. I would be more impressed if they had detected the structural relations that exist between the μ -phase and the precipitate phases in the Al 7xxx system. Yes, there are such relations.

Response: Thank you very much for mentioning this point. We have consulted relevant literatures of Al 7xxx system, and just as what you have mentioned, there are certain structural relations between μ -phase and the precipitate phases in the Al 7xxx system, as shown in the figure below. C14 Laves phase (also known as η -MgZn₂) is an important precipitated phase in Al 7xxx system alloy. Figure a is the atomic structure of the C14 phase observed along [110], and the two red rhombuses represent the unit cell of the C14 phase [1]. The part in the black dotted box (a rhombus and a parallelogram) equals to the two red rhombuses, which is another manifestation of the unit cell of this phase. Figure

b is the atomic structure of rhombohedral μ phase observed along $[1-10]$, and a rhombus and a rectangle represent the unit cell of this phase [2]. In many literatures [1, 3-7], we have found that the μ phase with unit cell form often exists in η phase of Al 7xxx system, as a form of defect. Figure c-e show several common defect structures of η phases containing μ phase in the Al 7xxx system alloys. Among them, the black dotted line is C14 Laves phase, and the red box is μ phase. The attachment part of the blue circles can be extracted as the structure in the blue box at the bottom of the figure c-e, i.e., the commonly mentioned flattened hexagonal structure in many literatures. The structure in Figure c has been reported in the Fig. 6 of the literature [1], the structure in Figure d has been reported in the Fig. 6 of the literature [3], and the structure in Figure e has been reported in the Fig. 6 of the literature [4]. The interface structures and orientation relationships of C14 and μ phase in these defect structures are the same, i.e., $(1-11)_{C14} // (001)_{\mu}$ and $[110]_{C14} // [1-10]_{\mu}$, and these orientation relationships also exist in other alloys [2, 8].

Reference

- [1] T.F. Chung, Y.L. Yang, M. Shiojiri, C.N. Hsiao, W.C. Li, C.S. Tsao, Z.S. Shi, J.G. Lin, J.R. Yang, An atomic scale structural investigation of nanometre-sized eta precipitates in the 7050 aluminium alloy, *Acta Mater.*, 174 (2019) 351-368.
- [2] Z.Q. Yang, L.F. Zhang, M.F. Chisholm, X.Z. Zhou, H.Q. Ye, S.J. Pennycook, Precipitation of binary quasicrystals along dislocations, *Nature Communications*, 9 (2018).
- [3] A. Bendo, T. Maeda, K. Matsuda, A. Lervik, R. Holmestad, C.D. Marioara, K. Nishimura, N. Nunomura, H. Toda, M. Yamaguchi, K. Ikeda, T. Homma, Characterisation of structural similarities of precipitates in Mg-Zn and Al-Zn-Mg alloys systems, *Philos. Mag.*, 99 (2019) 2619-2635.
- [4] T.F. Chung, Y.L. Yang, C.L. Tai, M. Shiojiri, C.N. Hsiao, C.S. Tsao, W.C. Li, Z.S. Shi, J.G. Lin, H.R. Chena, J.R. Yang, HR-STEM investigation of atomic lattice defects in different types of eta precipitates in creep-age forming Al-Zn-Mg-Cu aluminium alloy, *Mater. Sci. Eng. A-Struct. Mater. Prop. Microstruct. Process.*, 815 (2021).

- [5] T. Yang, Y. Kong, J.B. Lu, Z.J. Zhang, M.J. Yang, N. Yan, K. Li, Y. Du, Self-accommodated defect structures modifying the growth of Laves phase, *Journal Of Materials Science & Technology*, 62 (2021) 203-213.
- [6] C.D. Marioara, W. Lefebvre, S.J. Andersen, J. Friis, Atomic structure of hardening precipitates in an Al-Mg-Zn-Cu alloy determined by HAADF-STEM and first-principles calculations: relation to η -MgZn₂, *J. Mater. Sci.*, 48 (2013) 3638-3651.
- [7] Y. Zhang, M. Weyland, B. Milkereit, M. Reich, P.A. Rometsch, Precipitation of a new platelet phase during the quenching of an Al-Zn-Mg-Cu alloy, *Scientific Reports*, 6 (2016).
- [8] M. Slapakova, A. Zendegani, C.H. Liebscher, T. Hickel, J. Neugebauer, T. Hammerschmidt, A. Ormeci, J. Grin, G. Dehm, K.S. Kumar, F. Stein, Atomic scale configuration of planar defects in the Nb-rich C14 Laves phase NbFe₂, *Acta Mater.*, 183 (2020) 362-376.

Changes to manuscript: Following the reviewer's suggestion, we added several references, to supplement previously cited ones in "... include the coherent precipitate phases in aluminum alloys⁶⁻¹¹".

The statement in line 50 that these phases are more complex than Laves phases, I cannot fully agree on. Laves phases, although the structure of phases like Zn₂Mg are relatively simple, they do display complex stacking faults (as their precursors in the 7xxx system alloys clearly show), and they are structurally related through (disordered) FK phases to quasicrystals, which are what you would call 'more complex' than these phases.

Response: We thank the reviewer for raising this point. The 'complexity' we described here has considered the types of coordination polyhedra required to compose different TCP phases. Frank and Kasper found that a number of complex phases can be represented as packings of hard spheres of different radii and the basic stacking units are the coordination polyhedra [9-11]. The coordination polyhedra required by Laves phase includes Z12 and Z16 [11], while the coordination polyhedra required by μ and P phase includes Z12, Z14, Z15, and Z16. So in this paper, we stated that the latter is more complex than the former. We agree with the reviewer about the stacking faults. To avoid misunderstanding, we have deleted the 'are more complex than Laves phases and' in the revised version.

Reference

- [9] F.C. Frank, J.S. Kasper, Complex alloy structures regarded as sphere packings .1. definitions and basic principles, *Acta Crystallographica*, 11 (1958) 184-190.

[10] F.C. Frank, J.S. Kasper, Complex alloy structures regarded as sphere packing .2. analysis and classification of representative structures, Acta Crystallographica, 12 (1959) 483-499.

[11] B. Seiser, R. Drautz, D.G. Pettifor, TCP phase predictions in Ni-based superalloys: Structure maps revisited, Acta Mater., 59 (2011) 749-763.

Line 52 claims 'unusually flat and sharp interfaces'. What is 'unusual'? Are the stacking faults flatter and sharper than in diamond silicon? I think not. Neither Si nor stacking faults in Si are unusual. The claim is wrong.

Response: Thank you for your advice. We have modified this part in the revised version.

The figures and annotation are ok, except they could be simplified.

Response: Thank you for your advice. We have modified some figures and annotations in the revised version.

In short: This work has sound and interesting findings, well documented. The findings relate to how TCP phases in super-alloys connect structurally. The findings may be important, as these alloys are technologically materials of increasing importance. However, the presentation is very technical. Unless I misunderstand the directions of Nature Communications, my opinion is that this work fits better in specialist journals like Acta Materialia.

A final comment:

In Fig. 1e, the authors describe a motif as 'pentagonal' configurations. These are the same that exist with precipitates in the 7xxx system, and may be viewed as centred deformed icosahedra.

Response: Thank you for your advice. We have made supplement to this part, as follows:

“This can be explained by noting that pentagonal anti-prism (may be viewed as icosahedra³¹ in three-dimensional view) structures (Fig. 1d) are known to stabilize the μ structure³².”

Reviewer #2 (Remarks to the Author):

In this manuscript, the authors studied the atomic arrangement of the faulted twin boundary (FTB) in the μ phase and the intergrowth of P (or σ) phase with μ phase at the FTB by HRSTEM. It is proposed that the FTB in the μ phase sets up an easy pathway for the long-range diffusional transport of the smaller 3d transition metal species and the affected neighboring atoms rearrange to reconfigure the

lattice into that of the P (or σ) phase. This study provides useful information about the planar defects and the mechanism of the intergrowth of P (or σ) phase with μ phase. However, since the stacking fault energy can be significantly influenced by the element and composition, the energies of the planar defects calculated by using a simplified Co₇W₆ model are not convincing. The reason why the spatial location/distribution of the FTBs instead of thermodynamics determines the type and size of the new phase formed from the parent μ structure has not been properly discussed. The detailed comments and questions from the reviewer are as follows.

1. The authors reported that FTBs are abundant in the μ phase. According to literature [1] different types of planar defects on the $\{1\ -1\ 0\ 2\}$ plane were observed in μ phase. Do other types of non-basal planar defects exist in the μ phase in the present work? Are the FTBs the dominant planar defects in the sample? How are the FTBs distributed? Does the intergrowth of P (or σ) phase with μ phase only occur at the vicinity of the FTBs? The information is necessary for readers to understand why the authors correlated the intergrowth of P (or σ) phase with the FTBs rather than the other planar defects.

[1] S. Gao, Z.-Q. Liu, C.-F. Li, Y. Zhou, T. Jin, In situ TEM investigation on the precipitation behavior of μ phase in Ni-base single crystal superalloys, *Acta Materialia* 110 (2016) 268-275.

Response:

We thank the reviewer for these suggestions. First, for “According to literature [1] different types of planar defects on the $\{1\ -1\ 0\ 2\}$ plane were observed in μ phase” mentioned by the reviewer above and “However, in literature [2] experimental results show that there are various types of non-basal planar defects in Co₇W₆ but the FTB in the form of TS_{[001]-5} was not reported” mentioned in Question 2, we make the following interpretation based on the literature and our observations:

In the literature [1], two plane defects, i.e., incoherent twin and stacking fault, were observed in (110) plane of μ phase (corresponding to the (1-12) plane expressed in hexagonal crystal system in the literature [1, 2]). Among them, stacking fault is similar to the planar defect in (110) plane in the literature [2]. And for the first plane defect, incoherent twin, it is actually the FTB observed along $[1\ -1\ 0]$ in μ phase (see the figure a below), which is reported in our work. For the second defect — stacking fault, we have photographed similar structure in our sample, as shown in figure b. Figure c is the atomic diagram of this structure, from which we can see that when the spacing between two FTBs (orange dotted line) is D , a stacking fault of $1/2t_1$ (its magnitude is 0.23nm) seems to happen to the perfect structure on the right relative to that on the left, which is completely consistent with the $1/2\ t_1$ stacking fault observed in (110) plane in the literature [2]. Similarly, Figure d shows the defect structure of μ phase when the spacing between two FTB is $2D$. A stacking fault of $t_1=0.46\text{nm}$ seems to happen to the perfect structure on the right relative to that on the left, which is completely consistent with the stacking fault of t_1 observed in (110) plane in the literature [1]. It is therefore reasonable to speculate that the stacking faults reported in literature [1, 2] are these structures in figures c-d. It can be found that these

defect structures that look like stacking faults are actually FTBs. Perhaps due to the small spacing between the FTBs and the limited resolution of previous observation methods, they were simply considered as stacking faults in the past.

Based on the above, we provide the following reply to the questions raised by the reviewer. In our samples, except for the structure after transformation of FTB, only FTB planar defect can be observed in the (110) plane of the μ phase. The spacing and initial distribution of FTBs are random to some extent (seen in Fig. 1f and Extended Data Fig. 9b-c). These are also confirmed in Figs 2, 4 and 8 in literature [1]. Besides, in the vicinity of each intergrowth P (or σ) of μ phase, we all found FTB structure, and similarly, in the literature [3-5], (110) planar defects can be found in the vicinity of intergrowth P (or σ) of μ phase. Our current observations together with the above discussion regarding the (110) planar defect and the calculations of the most stable (110) defect structures of μ phases with different compositions in the next question all indicate that these (110) planar defects are most likely FTBs. All these indicate that there are close relations between intergrowth P (or σ) of μ phase and the (110) FTB of μ phase.

Reference

- [1] S. Gao, Z.Q. Liu, C.F. Li, Y.Z. Zhou, T. Jin, In situ TEM investigation on the precipitation behavior of mu phase in Ni-base single crystal superalloys, *Acta Mater.*, 110 (2016) 268-275.
- [2] P.A. Carvalho, J.T.M. De Hosson, Stacking faults in the Co7W6 isomorph of the mu phase, *Scr. Mater.*, 45 (2001) 333-340.
- [3] X.P. Tan, J.L. Liu, T. Jin, Z.Q. Hu, H.U. Hong, B.G. Choi, I.S. Kim, C.Y. Jo, Intergrowth of P phase with mu phase in a Ru-containing single-crystal Ni-based superalloy, *Philosophical Magazine Letters*, 92 (2012) 556-562.
- [4] D.S. Zhou, H.Q. Ye, K.H. Kuo, An hrem study of the intergrowth structures of sigma-related phases and the mu-phase, *Philosophical Magazine a-Physics of Condensed Matter Structure Defects and Mechanical Properties*, 57 (1988) 907-922.

[5] S. Gao, Z.Q. Liu, J.P. Cui, Y.Z. Zhou, T. Jin, In situ transformation from P phase to phase in rhenium-containing single-crystal superalloy during thermal exposure, *Philosophical Magazine Letters*, 97 (2017) 188-196.

Changes to manuscript: According to the reviewer's suggestions, we have made related statement of the relationship between FTB and the intergrowth P phase in the sixth paragraph of the main text, and made a supplemental description of the distribution of FTBs in the penultimate paragraph.

2.The authors used Co_7W_6 as a simplified model to calculate the interfacial energies of several non-basal defects and concluded that the FTBs in the form of $\text{TS}[001]-5$ are abundant because of the lowest energy. However, in literature [2] experimental results show that there are various types of non-basal planar defects in Co_7W_6 but the FTB in the form of $\text{TS}[001]-5$ was not reported. It indicates that at least the FTB in the form of $\text{TS}[001]-5$ is not dominant in Co_7W_6 . Therefore, it is not convincing to explain why such FTBs are abundant in the μ phase using the model of Co_7W_6 . Besides, such FTBs were not frequently observed in other superalloys having different compositions. Since the element and chemical composition can influence the stacking fault energy, the occurrence of this type of defect might be related to the composition of the sample studied in the present work. Have the authors considered the influence of the element and chemical composition on the defects?

[2] P.A. Carvalho, J.T.M. De Hosson, Stacking faults in the Co_7W_6 isomorph of the μ phase, *Scripta Materialia* 45(3) (2001) 333-340.

Response: We thank the reviewer for raising these questions. For 'in literature [2] experimental results show that there are various types of non-basal planar defects in Co_7W_6 but the FTB in the form of $\text{TS}_{[001]-5}$ was not reported' mentioned by the reviewer, we have made related explanation in the last question, i.e., (110) defect ((1-12) plane expressed in hexagonal crystal system) in literature [2] is the defect structure observed by us when the spacing between two FTBs is D. Besides, we agree with the reviewer's statement that 'the element and chemical composition can influence the stacking fault energy'. In order to analyze whether the stability of (110) FTB is universal, the interface energy calculations of the (110) defects of μ phase with three constituent components (Co_7Mo_6 , $\text{Co}_7\text{W}_3\text{Re}_3$, and $\text{Co}_7\text{W}_3\text{Mo}_3$) have been added, in addition to the original Co_7W_6 model. In Extended Data Fig. 6 (methods), we have compared the interfacial energies of 32 possible interface structures of the (110) defects of each type of μ phase. It can be found that the interface structures with the smallest energy in (110) defects of each type of μ phase are all FTB structure, i.e., $\text{TS}_{[001]-5}$ type. Hence, we think the stability of (110) FTB is universal, and element and chemical composition will make no change that it is more stable than other (110) defect structures. For 'such FTBs were not frequently observed in other superalloys having different compositions' mentioned by the reviewer, just as what we have mentioned in the Question 1, in fact, the existence of (110) defects in μ phase has been reported in many alloys with different compositions in the past [1-5]. However, due to the limited resolution of previous detection methods, the atomic structure of this defect was not reported and simulated carefully.

Changes to manuscript: We have added the related descriptions of the interface energy calculations of the (110) defects of μ phases with three constituent components in the fifth paragraph of the main text, Extended Data Fig. 6, and methods section.

3. The authors concluded that “it is the spatial location/distribution of the FTBs that determines the type and size of the new phase formed from the parent μ structure”. The P and σ phases are expected to have different thermodynamic stabilities at the heat treatment temperature, and thus the driving forces for the phase transformations from the parent μ to P and from μ to σ are expected to be different. How can the spatial location and distribution of the FTBs overwhelm the thermodynamics and control the phase transformations? Moreover, if the P and σ phases are thermodynamically more stable than the μ phase at the heat treatment temperature, they are expected to grow at the expense of the μ phase [3]. If the size of the new P and σ phases are also determined by the spatial location/distribution of the FTBs, how do the P and σ phases grow?

[3] X.P. Tan, J.L. Liu, T. Jin, Z.Q. Hu, H.U. Hong, B.G. Choi, I.S. Kim, C.Y. Jo, Intergrowth of P phase with μ phase in a Ru-containing single-crystal Ni-based superalloy, *Philosophical Magazine Letters* 92(10) (2012) 556-562.

Response: We thank the reviewer for raising these questions. The spatial location/distribution of the FTBs will not overwhelm the thermodynamics and control the phase transformations. Instead, they will be influenced and regulated by different thermodynamic stabilities of different phases under external conditions such as heat treatment, and the growths of the P and σ phases are also therefore regulated. We added Fig. 5 and corresponding text in the penultimate paragraph of the main manuscript to describe and discuss these questions, specifically as follows:

‘It can be seen from the above that the formation of regular new P (or σ) phase requires a specific spacing between FTBs in μ phase, while the initial distribution and spacing of FTB in μ phase are random to some extent (Fig. 1f and Supplementary Fig. 9b-c). We will discuss below how the new phase thickens through the short-range diffusion of local atoms at the FTBs. Figure 5a is the HAADF image captured at a specific moment in the thickening process of intergrowth P phase, which includes three zones (A, B, and C), representing different stages of transformation (propagating to the upper right along the orange arrow). The arrangement of atoms in region A (containing thicker and more P phases) is shown in Fig. 5b, where the conversion is over and all distortions have relaxed. The tip of the transformation zone is now reaching region B (transition zone), in which the lattice distortion strain is still discernible. At the upper right, the atoms in region C (containing thinner and fewer P Phases) are arranged as shown in Fig. 5c, and its transformation to region A, has not yet begun. Comparing the atomic arrangement of region A and region C, it can be found that, except for some atoms in the blue block layer (thickness of 4D) of region A, the arrangements of the atoms on both sides above and below are exactly the same as region C. Also, in terms of the atom numbers and the relative positions

between atoms, the blue block layer is consistent with the same block layer of region C. We can deduce from this sequence of events that the transformation from region C to region A, i.e., the thickening process of intergrowth P phase can be accomplished by the short-range diffusion (shuffle) of local atoms, and it is similar to a horizontal flip from the planar view. This transformation changed the distribution and spacing among FTBs (shown by black dotted line in Fig. 5b-c), and the number of FTBs with a spacing of $2D$ increased significantly after the transformation. Figure 4 shows that the regular P phase requires a spacing of $2D$ between FTBs, so the transformation from region C to region A thickens the P phase. The transformation process can be further simplified to Fig. 5d, and the atomic structures of region C and A are simplified as line 1 and line 2, respectively. Endpoints and turning points of line 1 and line 2 indicate the presence of the FTBs here. The local atomic shuffle performs a transformation from the middle red line to the orange line, and the transformation can change the location of FTBs (The FTB at number 5 transforms to perfect μ and an identical FTB appears at number 5'). When the same transformation happens to line 3 (Fig. 5e), which has similar structure to line 1, it can change the number of FTBs (from four to six). To illustrate, Fig. 5f is used to describe the growth process of the intergrowth P (or σ) phase. During the formation of the parent μ phase, there will be a lot of initial randomly-distributed (110) FTBs with low stacking fault energy, which is represented by the solid black line in schematic i), with a total number of m . When a new phase is more stable at a certain heat treatment temperature, via a similar local atomic short-range diffusion at FTBs like Fig. 5a, the number of FTBs n can proliferate to $3n$, as shown in schematic ii) of Fig. 5f, and their distribution and spacing can also be regulated as schematic iii) in Fig. 5f. The FTBs with certain spacing required for the growth of new phases can be thus obtained by this short-range diffusion processes, as shown in Fig. 4 and 5. The driving forces of this transformation are the difference in Gibbs free energy of these phases at the heat treatment temperature.'

Fig. 5 | The HAADF image and schematic illustration of the thickening process of intergrowth P phase. **a**, The HAADF image projection along $[1-10]_{\mu}$ showing the thickening process of intergrowth P phase. The orange arrow in **(a)** marks the direction of the progressive transformation. Regions A-C corresponds to the three sequential stages of transformation. **b-c**, Atomic models corresponding to the regions A and C in **(a)** respectively. The differences between the atomic arrangement in region A and that in region C are indicated by the blue block layer (thickness of $4D$) in **(b)**. The black dashed lines mark the interface position of the FTBs in **(b-c)** and the distances between two adjacent FTBs are added. The atomic structures of region C and A can also be simplified as line 1 and line 2, respectively. Their endpoints and turning points indicate the presence of the FTBs here, numbered by the green text in **(d)**. **d**, Schematic illustration of the location of FTBs changed by the transformation in **(a)**. The FTB at number 5 transforms to perfect μ and an identical FTB appears at number 5'. **e**, The schematic example of the number of FTBs changed by the same transformation. The structure of line 3 is similar to that of line 1, only different at the top. **f**, The brief schematic of the growth process of intergrowth P (or σ) phase in μ phase. The schematic i) in **(f)** shows that there are many randomly-distributed original (110) FTBs in the parent μ phase, represented by the solid black lines, with a total number of m . Via the local atomic short-range diffusion at FTBs, n FTBs can proliferate to be $3n$ as illustrated by the 1st and 2nd FTB shown in ii), and their distribution and spacing can also be regulated as illustrated by the i^{th} and nearby FTBs shown in iii). The new phase in schematic iii) is an example of the P phase (FTBs with a spacing of $2D$). If the new phase is σ phase, the spacing between FTBs is D .

Reviewer #3 (Remarks to the Author):

In their manuscript “Atomistic Mechanism of Phase Transformation between Topologically Close-packed Complex Intermetallics”, Jin et al. present an in-depth experimental and numerical study of the formation and transformation of topologically close-packed (TCP) phase. By focusing on Ni-base superalloy materials, the authors demonstrate the transformation of a μ phase into other TCP phases (P or σ) by means of the climb of a Frank partial dislocation.

The manuscript presents very precise atomic-scale experimental evidences of the formation and propagation of the Frank dislocation. I particularly appreciate the demonstration of the unconventional Burgers vector. The first principles calculations supporting the experiments are judiciously performed on a surrogate material as the complete superalloy remains obviously out of range of the capabilities of the DFT method. These calculations provide a strong support in building the defect structures presented in Fig. 2 and Fig. 4.

The mechanism evidenced by the authors is highly interesting for the community and is probably transferable to other class of material. As an example, it has been recently observed Frank partial dislocations in Ti-base MAX phases [Yu et al., Scripta Materialia 191 (2021) 34] and this defect could be associated with phase transformation by following a mechanism similar as the one proposed by this work. A short comment on the transferability into other TCP / complex phases of the mechanism evidenced by the authors could strengthens the discussion of the manuscript.

Response: We thank the reviewer for highlighting this reference. It provides an excellent example indicating that the findings of this paper are promising to be extended to other class of material to arouse wider interest. We added related statement and literature in the last paragraph of the main text, specifically as follows:

“Furthermore, the Frank partial dislocations similar to that in our work have recently been reported³³ in MAX phase materials, which may play an important role in the field of self-lubrication, oxidation-resistant coatings, self-healing and energy materials. These defects were found to have close relations with the phase transformation between the materials. Our work is expected to promote the understanding of structural transformations within these and other similarly complex materials.”

REVIEWERS' COMMENTS

Reviewer #1 (Remarks to the Author):

A far as I can see, the authors have done a good job, met all objections, responded well to the objections I had. I am very happy with the improvements in the quality, and have no further comments.

Reviewer #2 (Remarks to the Author):

The authors have addressed my comments and questions. They have shown the universal stability of the FTBs in different μ phases and the role of the FTBs in the phase transformation and thickening process with new and solid results. These results improve our understanding of the phase transformation between the TCP phases. Finally, as most of the references cited in this manuscript use the hexagonal description of the μ phase, I would suggest using hexagonal description in this manuscript or at least adding the hexagonal descriptions of the crystallographic planes and directions to the figures and tables. It would be much easier for the readers to read and compare the present work with literature.

Reviewer #3 (Remarks to the Author):

The authors have provided careful and precise replies to the comments of all three reviewers. The additional figure and paragraph in the discussion are especially useful. To my opinion, the concerns previously raised have been fully addressed.